# Disorder-enabled Andreev reflection of a quantum Hall edge

Vladislav D. Kurilovich ⓘ[1] ✉, Zachary M. Raines[1] & Leonid I. Glazman ⓘ[1]

We develop a theory of charge transport along the quantum Hall edge proximitized by a superconductor. We note that generically Andreev reflection of an edge state is suppressed if translation invariance along the edge is preserved. Disorder in a "dirty" superconductor enables the Andreev reflection but makes it random. As a result, the conductance of a proximitized segment is a stochastic quantity with giant sign-alternating fluctuations and zero average. We find the statistical distribution of the conductance and its dependence on electron density, magnetic field, and temperature. Our theory provides an explanation of a recent experiment with a proximitized edge state.

Recent interest in engineering an exotic superconductor have renewed the effort to combine the superconducting proximity effect with a quantizing magnetic field. The combination of the two has been proposed as a route to realize new quasiparticles, such as parafermions[1,2], which may be employed for topological quantum computing[3].

The picture of the proximity effect is based on Andreev reflection, in which an electron incident on the interface between a normal-state conductor and a superconductor is reflected as a hole[4]. In fact, this electron-hole conversion has been demonstrated[5,6] in focusing experiments utilizing a weak magnetic field $B$ to bend the electron and hole trajectories. Classically, trajectory bending due to the Lorentz force leads to formation of skipping orbits propagating along the boundaries. At fixed energy, quasiclassical quantization results in a discrete spectrum of angles $\alpha_n(B)$ such a trajectory may form with the boundary. For electron-hole conversion at a boundary with a clean superconductor, the angles of incidence and reflection obey the retroreflection condition, $\alpha_n(B) + \alpha_m(B) = \pi$. As follows from a simple geometric analysis, this requires the centers of electron and hole cyclotron orbits to be mirror-symmetric with respect to the interface, $y_{c,n}(B) + y_{c,m}(B) = 0$. In fact, the centers of orbits are integrals of motion and the above symmetry condition is equivalent to the conservation of momentum component parallel to the interface. Therefore, the symmetry condition is exact and valid in a fully-quantum description, beyond the semiclassical approximation.

In the conditions of the quantum Hall effect, $y_c$ can be viewed as the positions of edge states. Their number decreases with the increase of the magnetic field $B$. In high field (that is, at filling factor $v = 2$), a single edge state remains, $n = m = 1$. Application of the mirror-symmetry condition shows that the proximitization is effective only when $y_{c,1}(B) = 0$. This configuration is realized at a single value of $B$. Appreciable electron-hole conversion occurs only at that fine-tuned value of the field (see Supplementary Note 1 for further discussion).

Disorder, however, breaks the momentum conservation and relaxes the mirror-symmetry constraint. This allows for an appreciable electron-hole conversion at any magnetic field. Indeed, a strong conversion signal was observed in recent experiments[7–10] without fine-tuning; the need of high critical fields $H_{c2}$ dictated the use of disordered ("dirty") superconductors. Robust Andreev reflection, being enabled by disorder, is naturally sensitive to its realization in a sample. As a result, the charge transport varies stochastically with control parameters such as the magnetic field or the electron density, as is observed both in experiment[8] and in numerical simulation[11].

The crucial difference of conduction along the proximitized quantum Hall edge from the conventional mesoscopic transport stems from the chirality of the edge states. This renders the well-established theory of mesoscopic conductance fluctuations[12,13] inapplicable. In this work, we develop a quantitative theory of mesoscopic quantum transport along the proximitized chiral edge, making predictions for the statistics of conductance fluctuations and their dependence on electron density, magnetic field, and temperature. The results obtained for chiral transport differ substantially from their counterpart in usual conductors.

[1]Department of Physics, Yale University, New Haven, CT 06520, USA. ✉e-mail: vlad.kurilovich@yale.edu

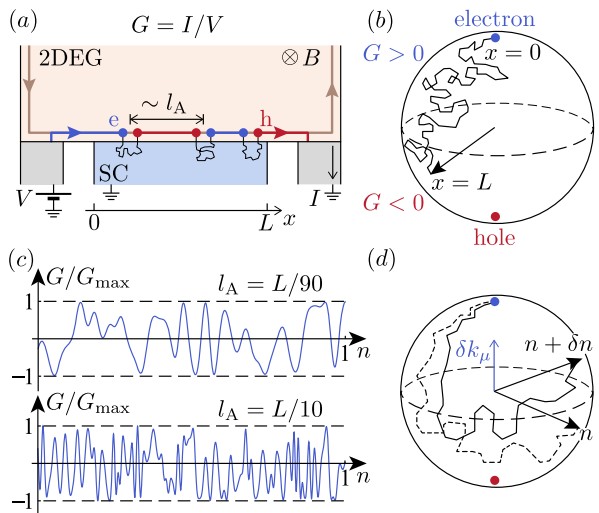

**Fig. 1 | Transport along the quantum Hall edge proximitized by a disordered superconductor. a** Electrons are launched toward the proximitized segment from an upstream electrode biased by voltage $V$. An electron propagating along the segment converts randomly into a hole over the distance $l_A$, which is controlled by disorder in the superconductor, see Eq. (8). **b** Evolution of the electronic wave function, see Eq. (10), is similar to the motion of a "spin" in a stochastic effective "magnetic field". The conductance $G = I/V$ is determined by the result of a random walk of a point on a Bloch sphere. **c** $G$ is a random quantity that fluctuates upon varying the electron density $n$ in the 2DEG (traces are simulated using Eq. (10); units of $n$ are the same for the two plots and are otherwise arbitrary). **d** The loss of correlation between the values of $G$ upon a change in $n$ is quantified by function $\mathcal{C}(\delta n)$, see Eqs. (15)–(17). The origin of the correlations loss is illustrated by the divergence between two stochastic trajectories on a Bloch sphere. The "spins" corresponding to different values of $n$ experience a different effective "magnetic field", and thus drift apart in the course of evolution. The separation of the "spins" is slower for stronger disorder. As the result, the trace $G(n)$ in (**c**) is smoother for smaller $l_A$.

## Results
### Model
We are interested in the linear conductance $G$ in a three-terminal setting, see Fig. 1(a). To find $G$, we start with the Hamiltonian

$$H = H_{2DEG} + H_{SC} + H_T. \tag{1}$$

Here, $H_{2DEG}$ describes the two-dimensional electron gas (2DEG) in a $\nu = 2$ quantum Hall state. $H_{SC}$ is the Hamiltonian of the superconductor. We consider the experimentally relevant[7–10] "dirty" limit $l_{mfp} \ll \xi$, where $l_{mfp}$ and $\xi$ are, respectively, the electron mean free path and the coherence length in the superconductor. Coupling between the 2DEG and superconductor is described by the tunneling Hamiltonian[14,15]

$$H_T = t \sum_\sigma \int_0^L dx (\partial_y \psi_\sigma^\dagger(x,0) \partial_y \chi_\sigma(x,0,0) + \text{h.c.}), \tag{2}$$

where $\psi_\sigma(x,y)$ and $\chi_\sigma(x,y,z)$ are annihilation operators for an electron with spin $\sigma = \uparrow$ or $\downarrow$ in the 2DEG and superconductor, respectively. The interface of length $L$ is located at $y = z = 0$; $\partial_y$ denotes the partial derivative in the normal to the interface direction. For simplicity, we assume that the tunneling amplitude $t$ is uniform along the interface.

For the purpose of describing transport at low temperature and bias, it is convenient to derive an effective Hamiltonian focusing on chiral electrons at the 2DEG's edge,

$$H_{eff} = H_{edge} + H_{prox}. \tag{3}$$

The first term is obtained by projecting $H_{2DEG}$ onto the subspace of edge states belonging to a single Landau level

$$H_{edge} = \sum_\sigma \int dx\, \eta_\sigma^\dagger(x) \hbar v[-i\partial_x - k_\mu]\eta_\sigma(x). \tag{4}$$

Here, $\eta_\sigma(x)$ is a field operator for chiral electrons with $\sigma = \uparrow$ or $\downarrow$, $v$ is their velocity, and $k_\mu$ is the Fermi momentum; we neglect the Zeeman splitting. The second term in Eq. (3) describes the effect of superconducting proximity. It is obtained by a standard Schrieffer-Wolff transformation[16] that removes coupling (2) to the first order in $t$. For electron energies $E \ll \Delta$, as measured from the Fermi level, the transformation results in

$$H_{prox} = (\partial_y \Phi)^2 t^2 \int_0^L dx_1 dx_2\, \hat\eta^\dagger(x_1) \partial_{y_1 y_2}^2 \mathcal{G}(x_1, x_2) \hat\eta(x_2), \tag{5}$$

where $\hat\eta(x) = (\eta_\uparrow(x), -\eta_\downarrow^\dagger(x))^T$, the $2 \times 2$ matrix $\mathcal{G}(x_1, x_2)$ is the Green's function of the superconductor at $E = 0$ ($\partial_{y_1 y_2}^2$ is a mixed partial derivative with respect to $y_1$ and $y_2$; arguments $y_{1,2}, z_{1,2} = 0$ are suppressed for brevity), $\Delta$ is the energy gap in the superconductor, and $\Phi(y)$ is the transverse component of the edge state's wave function at the Fermi level.

Conductance $G$ at $T = 0$ can be expressed in terms of transmission amplitudes across the proximitized segment in the normal ($A_e$) and Andreev ($A_h$) channels at $E = 0$,

$$G = G_Q(|A_e|^2 - |A_h|^2), \tag{6}$$

where $G_Q = 2e^2/h$ is the conductance quantum. To find $G$ in the setup of Fig. 1(a), we thus need to solve a quantum-mechanical scattering problem.

### Andreev amplitude for a short segment
An electron experiences at most one Andreev reflection while propagating along a sufficiently short proximitized segment. The corresponding Andreev amplitude can be found perturbatively in $H_{prox}$. With the help of Born approximation, we obtain

$$A_h = -\frac{(\partial_y \Phi)^2 t^2}{\hbar v} \int dx_1 dx_2\, e^{ik_\mu(x_1 + x_2)} \partial_{y_1 y_2}^2 \mathcal{G}_{he}(x_1, x_2), \tag{7}$$

where $\mathcal{G}_{he}$ is the anomalous component of the superconductor Green's function[17].

The Green's function in Eq. (7) is determined by the interference of electron waves in the superconductor. The stochastic interference pattern is sensitive to a particular disorder landscape in the region of size $\sim \xi$ adjacent to the interface. Thus, $\mathcal{G}_{he}$ and $A_h$ of Eq. (7) are random quantities. The latter fluctuates upon varying the magnetic field or the electron density in the 2DEG.

To characterize the statistical properties of the amplitude, we first find $\langle A_h \rangle$. The averaging here is performed over a sufficiently broad window of magnetic fields or electron densities. Formally, it is equivalent to averaging over the possible disorder configurations in the superconductor[18]. With the help of the latter, more practical definition we obtain: $\langle A_h \rangle \propto \int dx_1 dx_2 e^{ik_\mu(x_1+x_2)} \langle \mathcal{G}_{he}(x_1 - x_2) \rangle \propto \int dx e^{2ik_\mu x} \propto \delta(k_\mu)$. We see that $\langle A_h \rangle = 0$ unless $k_\mu = 0$. In the following, we disregard such a fine-tuning and take $\langle A_h \rangle = 0$.

Next, we compute the average probability of the Andreev reflection $\langle |A_h|^2 \rangle$. As follows from Eq. (7), we need to average product of the anomalous Green's functions of the superconductor. Such an average can be expressed in terms of the normal-state diffuson and Cooperon via a standard procedure (see, e.g., Ref. 19). Assuming that the thickness of the superconducting film and $L$ exceed $\xi$, we obtain (see

Methods section for a detailed derivation):

$$\langle |A_{\mathrm{h}}|^2 \rangle = \frac{L}{l_{\mathrm{A}}}, \quad \frac{1}{l_{\mathrm{A}}} = \frac{4\pi g^2}{G_Q \sigma} \ln \frac{\xi}{l_{\mathrm{mfp}}}. \tag{8}$$

Here $g = 2\pi^2 G_Q t^2 (\partial_y \Phi)^2 \nu_{\mathrm{QH}} \nu_{\mathrm{M}} p_F / \hbar$ is the conductance per unit length of the interface between the quantum Hall edge and the metal in the normal state. Along with the dependence on $\Phi(y)$, the conductance $g$ is proportional to the one-dimensional density of edge states $\nu_{\mathrm{QH}} = 1/(2\pi\hbar v)$. It is also proportional to the normal-state density of states $\nu_{\mathrm{M}}$ and Fermi momentum $p_F$ in the superconductor. Unlike in the clean case, the leading contribution to the Andreev reflection comes from electron trajectories much longer than the Fermi wave length, with length scale set instead by $\xi \gg l_{\mathrm{mfp}}$. The presence of the logarithmic factor and the appearance of the normal-state conductivity $\sigma$ in $1/l_{\mathrm{A}}$ results from the diffusive motion of electron in the superconductor.

The perturbative result, Eq. (7), is applicable at $L \ll l_{\mathrm{A}}$. Under this condition, $A_{\mathrm{h}}$ is a Gaussian random variable which allows one to compute all moments of $A_{\mathrm{h}}$ distribution. Using Eq. (6) we find $\langle G \rangle = G_Q (1 - 2L/l_{\mathrm{A}})$ and $\langle\langle G^2 \rangle\rangle = \langle G^2 \rangle - \langle G \rangle^2 = 4 G_Q^2 L^2 / l_{\mathrm{A}}^2$ for the average value and fluctuation of the conductance.

## Conductance of a long segment

At $L \gg l_{\mathrm{A}}$, an incident electron experiences multiple Andreev reflections upon traversing the proximitized segment. The first-order perturbation theory cannot be applied directly to find the amplitude $A_{\mathrm{h}}$ in this case. Instead, we track how the quasiparticle wave function evolves along the segment piece by piece.

We break the segment into a series of short elements with length $\delta L$ satisfying $\xi \ll \delta L \ll l_{\mathrm{A}}$. Under these conditions, the Andreev amplitudes of different elements $\delta A_{\mathrm{h}}(x)$ are statistically independent and may still be evaluated perturbatively, $\delta A_{\mathrm{h}}(x) = \alpha(x) \cdot \sqrt{\delta L}$. In addition to Andreev reflections, a quasiparticle may experience forward scattering due to an excursion in the superconductor. Similarly to $\delta A_{\mathrm{h}}(x)$, we find for the electron forward scattering phase $\delta\Theta(x) = \vartheta(x) \cdot \sqrt{\delta L}$ (see Supplementary Note 2 for details of the derivation). Variables $\alpha(x)$ and $\vartheta(x)$ are Gaussian and independent, $\langle \alpha(x)\vartheta(x') \rangle = 0$. Using Eq. (8) and a similar relation for $\langle \Theta^2 \rangle$ we obtain for the correlators

$$\langle \alpha(x)\alpha^\star(x') \rangle = \langle \vartheta(x)\vartheta(x') \rangle = \frac{1}{l_{\mathrm{A}}} \delta(x - x'). \tag{9}$$

The change of the wave function across each element is small. Therefore, we can describe the wave function evolution by a differential equation:

$$i \frac{\partial}{\partial x} \begin{pmatrix} a_{\mathrm{e}}(x) \\ a_{\mathrm{h}}(x) \end{pmatrix} = \begin{pmatrix} -\vartheta(x) & \alpha^\star(x) \\ \alpha(x) & \vartheta(x) \end{pmatrix} \begin{pmatrix} a_{\mathrm{e}}(x) \\ a_{\mathrm{h}}(x) \end{pmatrix}. \tag{10}$$

Here $a_{\mathrm{e}}(x)$ and $a_{\mathrm{h}}(x)$ are the electron and hole components of the quasiparticle wave function, respectively (we also promoted $\alpha(x)$ and $\vartheta(x)$ from the variables defined on a discrete set of elements to the continuous fields).

Equation (10) describes a unitary evolution of a two-component spinor, which can be visualized as a random walk of a point on a Bloch sphere, see Fig. 1(b). We parameterize $a_{\mathrm{e}}(x) = \cos(\theta(x)/2)$ and $a_{\mathrm{h}}(x) = e^{i\phi(x)} \sin(\theta(x)/2)$, where $\theta$ and $\phi$ are polar and azimuthal angles on the sphere, respectively (in the parameterization, we suppressed the common phase as it is inconsequential for $G$). The conductance $G = G_Q \cos \theta(L)$ can be expressed in terms of a solution of Eq. (10) with initial condition $\theta(0) = 0$.

To determine the statistics of conductance fluctuations, we derive a Fokker-Planck equation[20] for the distribution function $\mathcal{P}(\theta,\phi|x)$ with

help of Eq. (9):

$$\frac{\partial \mathcal{P}}{\partial x} = \frac{1}{l_{\mathrm{A}}} \left( \Delta_{\theta,\phi} + \partial_\phi^2 \right) \mathcal{P}. \tag{11}$$

Here $\Delta_{\theta,\phi}$ is the Laplace-Beltrami operator. Parameter $1/l_{\mathrm{A}}$ plays the role of a diffusion coefficient in the amplitude's random walk. Equation (11) can solved straightforwardly in terms of angular harmonics, $\mathcal{P}(\theta,\phi|x) = \sum_{l=0}^\infty (2l+1) P_l(\cos\theta) e^{-l(l+1)x/l_{\mathrm{A}}}/4\pi$, where $P_l(z)$ are Legendre polynomials. The independence of $\mathcal{P}(\theta,\phi|x)$ on $\phi$ stems from the azimuthal symmetry of Eq. (11) and its initial condition.

Using the found distribution function, we obtain for the average conductance:

$$\langle G \rangle = G_Q e^{-2L/l_{\mathrm{A}}}. \tag{12}$$

At $L \gg l_{\mathrm{A}}$, conductance $G$ is distributed uniformly in the interval $[-G_Q, G_Q]$ with $\langle G \rangle = 0$ and variance $\langle\langle G^2 \rangle\rangle = G_Q^2/3$. Thus, the conductance fluctuations pattern is sign-alternating and evenly distributed between positive and negative values, see Fig. 1(c). While the exponential with $L$ decay similar to our Eq. (12) was also demonstrated in related setups in Refs. 21,22, these works missed the giant fluctuations of conductance.

## Suppression of fluctuations by vortices

Only a type II superconductor can withstand magnetic field $B$ required to enter the quantum Hall regime in the 2DEG. Such field induces vortices, which lead to a non-vanishing density of states in the superconductor at the Fermi level[23]. As a result, an electron or a hole propagating along the edge can tunnel normally into the superconducting electrode thus not contributing to $G$. This leads to attenuation of conductance fluctuations.

The probability of an incident electron to survive the propagation along the proximitized segment and reach the downstream electrode (as a particle or as a hole) decreases exponentially with $L$:

$$p_{\mathrm{surv}} = \exp(-\gamma L), \quad \gamma = \frac{g}{G_Q} \frac{\bar{\nu}}{\nu_{\mathrm{M}}}. \tag{13}$$

Here $\gamma$ is the probability of normal tunneling into superconductor per unit length of the segment. It is determined by the induced by vortices density of states $\bar{\nu}$ taken at $E = 0$ and averaged along the interface. Despite the attenuation, at $L \gg l_{\mathrm{A}}$ the conductance distribution remains uniform. However, its spread reduces to the interval $[-G_{\mathrm{max}}, G_{\mathrm{max}}]$ and its variance becomes

$$\langle\langle G^2 \rangle\rangle = \frac{G_{\mathrm{max}}^2}{3}, \quad G_{\mathrm{max}} = G_Q p_{\mathrm{surv}}. \tag{14}$$

Ratio $\bar{\nu}/\nu_{\mathrm{M}}$ in Eq. (13) increases with $B/H_{c2}$, reaching unity at the upper critical field, $B = H_{c2}$. Consequently, $\langle\langle G^2 \rangle\rangle$ decreases with increasing $B$. This is qualitatively consistent with the observations of Ref. 8.

## Conductance correlation function

We now find the correlation function of the conductance fluctuations with the electron density $n$ in the 2DEG,

$$\mathcal{C}(\delta n) = \langle\langle G(n) \cdot G(n + \delta n) \rangle\rangle. \tag{15}$$

Variation of density $\delta n$ shifts the Fermi momentum of chiral electrons by $\delta k_\mu = \delta n (\partial\mu/\partial n)/(\hbar v)$, where $\partial\mu/\partial n$ is the inverse compressibility of the quantum Hall state. $\delta k_\mu$ affects the phases of Andreev reflection amplitudes, whose interference determines the conductance. We see from Eq. (7) that $\alpha(x) \to \alpha(x) e^{2i\delta k_\mu x}$ upon changing $n \to n + \delta n$. Applying this modification to Eq. (10) and using Eq. (9), we derived a differential

equation for $\mathcal{C}(\delta n)$ as a function of $L$ (see Supplementary Note 3). Solving it, we find at $L \gg l_A$:

$$\mathcal{C}(\delta n) = \langle\langle G^2 \rangle\rangle \, \exp\left[-\frac{4}{3}\left(\frac{\delta n}{n_{cor}}\right)^2\right]. \tag{16}$$

The correlation density $n_{cor}$ is given by:

$$n_{cor} = \frac{\partial n}{\partial \mu}\frac{\hbar v}{\sqrt{l_A L}}. \tag{17}$$

The dependence of Eq. (17) on $L$ and $l_A$ is of particular note. Firstly, $n_{cor} \propto 1/\sqrt{L}$ reflects the diffusive character of the wave function evolution. In contrast, periodic oscillations of the quasiparticle between electron and hole states in the absence of disorder would lead to $\mathcal{C}(\delta n)$ variation on a scale $\delta n \propto 1/L$[24]. Secondly, $n_{cor} \propto 1/\sqrt{l_A}$ increases with disorder in superconductor, as $l_A \propto \sigma$, cf. Eq. (8). Thus, the pattern of mesoscopic fluctuations is *smoother* for a dirtier superconductor, see Fig. 1(c). This unusual behavior is similar in its origin to the motional narrowing in nuclear magnetic resonance[25].

The conductance also fluctuates with the magnetic field. The generalization of Eq. (16) reads $\mathcal{C}(\delta n, \delta B) = \langle\langle G^2 \rangle\rangle \exp[-\frac{4}{3}\delta k_\mu^2 l_A L] \exp[-\frac{8}{3}(\delta g/g)^2 L/l_A]$. Change in the Fermi momentum $\delta k_\mu(\delta n, \delta B)$ varies the phases of the Andreev reflection amplitudes (as discussed above). Variation $\delta g(\delta n, \delta B)$ affects the amplitudes magnitude through the dependence of $\Phi(y)$ and $v$ on $B$ and $n$, cf. Eq. (7). Functions $\delta g$ and $\delta k_\mu$ acquire a particularly simple form in the limit of a small disorder-induced broadening of Landau levels, $\delta\varepsilon \ll \hbar\omega_c$ (here $\omega_c$ is the cyclotron frequency). We find $\delta g/g = \delta B/B$ and $\delta k_\mu(\delta n, \delta B) = \frac{1}{v}\frac{\partial\mu}{\partial n}[\delta n - v\delta B/\phi_0]$, where $v(n, B)$ is the quantum Hall filling factor and $\phi_0 = hc/e$ (see Supplementary Note 3 for details of the derivation). In expression for $\delta k_\mu(\delta n, \delta B)$, we also assumed the London penetration depth $\lambda \gg l_A$ to neglect the diamagnetic current effect.

## Effect of a vortex entrance

In the above we disregarded the entrance of vortices in the superconductor through the interface. An entering vortex introduces a kink in the phase of the order parameter near the interface. This affects the interference between the Andreev reflection processes thus leading to a jump $\delta G$ in the conductance.

The magnitude of $\delta G$ is a random quantity whose statistical properties depend on the relation between $d$ and $l_A$, where $d$ is the distance of the vortex core to the interface. We compute the variance, $\mathcal{C}_{jump}(d) = \langle(\delta G)^2\rangle$, where the average is evaluated over a window of electron densities of width exceeding $n_{cor}$. To do that, we compare the results of the wave function evolution along the proximitized segment before and after the vortex has entered.

In treating the entrance of a new vortex, we assume the regime of strong pinning, and thus neglect the shifts in the positions of other vortices. In these conditions, the vortex entrance leads to $\alpha(x) \to \alpha(x)e^{-i\delta\varphi(x-x_v)}$ in Eq. (10). Here, the phase $\delta\varphi(x - x_v) = \pi + 2\arctan([x - x_v]/d)$ interpolates between 0 and $2\pi$ over the interval $|x - x_v| \sim d$, where $x_v$ is the $x$-coordinate of the vortex core. The overall interference pattern does not change substantially if $d \ll l_A$. Under this condition, the conductance jump is small. It is also small in the opposite limit, $d \gg l_A$, in which the presence of $\delta\varphi(x - x_v)$ can be accounted for with the help of the adiabatic approximation applied to Eq. (10). We find (see Supplementary Note 4 for details of the derivation):

$$\frac{\mathcal{C}_{jump}(d)}{\langle\langle G^2 \rangle\rangle} = \begin{cases} \frac{32\pi d}{3 l_A}, & d \ll l_A, \\ \frac{4\pi l_A}{3d}, & d \gg l_A. \end{cases} \tag{18}$$

The two asymptotes match each other at $d \sim l_A$. In this case, the conductance jump is maximal and comparable to the signal itself,

$\mathcal{C}_{jump}(d) \sim \langle\langle G^2 \rangle\rangle$. This regime is relevant for the data presented in Ref. 8.

## Conductance fluctuations at finite temperature

In a conventional mesoscopic conductor, the electron transmission amplitudes at energies $E_1$ and $E_2$ are uncorrelated if $|E_1 - E_2| \gtrsim E_{Th}$. The Thouless energy here is determined by the electron propagation time across the sample; $E_{Th} = \hbar v/L$ in the ballistic limit. Thus, the ordinary mesoscopic conductance fluctuations[12,13] are smeared out at temperature $T \gtrsim T_{sm} = \hbar v/L$.

While quasiparticles propagate ballistically along the proximitized quantum Hall edge, the energy scale $\hbar v/L$ is *irrelevant* for the correlation of Andreev amplitudes. The main mechanism responsible for the variation of $A_h$ with $E$ is the dependence of the anomalous Green's function on $E/\Delta$ in Eq. (7) generalized to finite energy (we assume $\Delta \ll \hbar\omega_c$ and disregard other mechanisms which are controlled by $E/(\hbar\omega_c)$). Due to this dependence, the size of each step in the amplitude's random walk [cf. Eq. (10)] changes by a relative amount $\sim E^2/\Delta^2$ from its $E = 0$ value. This leads to the divergence of trajectories corresponding to energies $E_1$ and $E_2$ on the Bloch sphere. The fluctuations of $G$ are smeared out above $T_{sm}$ such that the angular separation between the trajectories end-points is $\delta\theta \sim 1$ for $|E_1 - E_2|, E_1 \sim T_{sm}$. We estimate $(\delta\theta)^2 \sim \frac{(E_1^2 - E_2^2)}{\Delta^4}\frac{L}{l_A}$, and thus find $T_{sm} \sim \Delta(l_A/L)^{1/4}$. The dependence of $T_{sm}$ on $L$ is in stark contrast with a conventional ballistic conductor result. The difference stems from the chiral nature of the edge, which prohibits backscattering and formation of standing waves.

The found weak dependence, $T_{sm} \propto L^{-1/4}$, prompts us to explore inelastic scattering as a mechanism of the fluctuations suppression. In one dimension, inelastic pair collisions are forbidden by the energy and momentum conservation[26]. Violation of translation invariance by disorder allows for the pair collisions at the edge and leads to a standard Fermi liquid estimate for the scattering rate[27], $\tau_{in}^{-1}(T) = b\,T^2$. The conductance fluctuations are suppressed at temperature exceeding $T_{in}$ such that $v\,\tau_{in}(T_{in}) \sim L$. We then find $T_{in} \propto L^{-1/2}$. The comparison of $T_{in}$ and $T_{sm}$ is sensitive to a coefficient $b$ which is not universal and depends on disorder (see Supplementary Note 5).

## Discussion

In summary, disorder allows for efficient Andreev reflection of a quantum Hall edge without fine-tuning, but it introduces randomness in the edge transport. Electrons stochastically convert into holes over a length scale $l_A$, see Eq. (8). This stochasticity results in conductance fluctuations with the variation of electron density or magnetic field strength. For a long edge, $L \gg l_A$, the average conductance $\langle G \rangle$ vanishes, see Eq. (12), while in the absence of vortices the individual realizations of $G$ vary within an interval $\pm 2e^2/h$. Electron tunneling into the cores of the vortices in the superconductor shrinks this interval, see Eqs. (13) and (14), due to electrons being lost to ground. The ensemble averaging of $G$ can be experimentally achieved in a given sample by varying the electron density $n$ by amount exceeding $n_{cor}$ of Eq. (17). At smaller variation, the values of $G$ are correlated, see Eq. (16). Variation of magnetic field also results in conductance fluctuations, including abrupt changes associated with a vortex entering the superconductor, see Eq. (18). At a finite temperature, thermal smearing and inelastic scattering suppress conductance fluctuations. The chiral nature of edge states, however, weakens the suppression compared to the case of conventional conductors.

We derived the above results for a single edge state, $v = 2$. However, they can be readily extended to the case of $v > 2$. The conductance of a long edge remains a random quantity with $\langle G \rangle = 0$ and a symmetric about zero distribution function. Using the random matrix theory, we can estimate the conductance variance as $\langle\langle G^2 \rangle\rangle \sim G_Q^2$ (in the absence of vortices). The independence of $\langle\langle G^2 \rangle\rangle$ of the number of

conduction channels is in the spirit of universal conductance fluctuations[28,29].

Our work uncovers the crucial role of disorder in inducing superconductivity in quantum Hall edge states. It explains the basic findings of experiment[8] including the observation of random conductance, with zero average. The quantum Hall effect requires application of a high magnetic field. Disorder is needed not only to facilitate proximity, but also to make the coherence length short, thus allowing a superconductor to withstand the high field. Therefore, understanding the effect of disorder is vital for assessing the prospects of engineering topological superconductors by proximitizing counter-propagating edge states[1,2,7,9].

## Methods
### Derivation of $1/l_A$

In this section, we present a detailed derivation of Eq. (8). For calculations, it is convenient to choose a gauge in which the vector potential vanishes at the interface between the superconductor and the 2DEG. In this gauge, the wave vector $k_\mu$ in the expression for the Andreev amplitude [see Eq. (7)] is related to the position of the edge state $y_c$, $k_\mu = y_c/l_B^2$, where $l_B = \sqrt{\hbar c/eB}$ is the magnetic length. At $\nu = 2$, we can estimate $k_\mu \lesssim 1/l_B$.

In the derivation of $1/l_A$, we dispense with the effect of the magnetic field in the superconductor. This is justified in sufficiently small fields, $B \ll H_{c2}$, where $H_{c2}$ is the upper critical field. Indeed, we will see that $1/l_A$ is determined by processes in which the quasiparticle propagates over a distance $\sim \xi$ within the superconductor. The magnetic field affects such processes leading to additional phase factors in their amplitudes. The corresponding phases can be estimated as $\sim B\xi^2/\phi_0 \sim B/H_{c2}$, where $\phi_0$ is the flux quantum. We see that the phases are small for fields $B \ll H_{c2}$, and can be neglected.

Dispensing with the effect of the field, we describe the superconductor with the standard BCS Hamiltonian:

$$H_{SC} = \sum_\sigma \int d^3r \, \chi_\sigma^\dagger(\boldsymbol{r}) \left[ -\frac{\hbar^2 \partial_{\boldsymbol{r}}^2}{2m} - \mu + U(\boldsymbol{r}) \right] \chi_\sigma(\boldsymbol{r})$$
$$+ \int d^3r \, \Delta \left( \chi_\uparrow^\dagger(\boldsymbol{r}) \chi_\downarrow^\dagger(\boldsymbol{r}) + \chi_\downarrow(\boldsymbol{r}) \chi_\uparrow(\boldsymbol{r}) \right). \quad (19)$$

Here $\chi_\sigma(\boldsymbol{r})$ is an annihilation operator for an electron with spin $\sigma$, $m$ is the effective mass, $\mu$ is the chemical potential, and $\Delta$ is the superconducting order parameter. $U(\boldsymbol{r})$ is the disorder potential, which we assume to be a Gaussian random variable with a short-ranged correlation function,

$$\langle U(\boldsymbol{r}) U(\boldsymbol{r}') \rangle = \frac{\hbar}{2\pi \nu_M \tau_{mfp}} \delta(\boldsymbol{r} - \boldsymbol{r}'). \quad (20)$$

We parameterized the correlation function by the normal-state density of states in the metal $\nu_M$ and the electron mean free time $\tau_{mfp}$. We assume that the superconductor is "dirty", $\Delta \cdot \tau_{mfp}/\hbar \ll 1$.

Let us now compute the average probability of the Andreev reflection (our approach is similar in spirit to that in Ref. 19). Using Eq. (7), we first represent $\langle |A_h|^2 \rangle$ as

$$\langle |A_h|^2 \rangle = \frac{(\partial_y \Phi)^4 t^4}{\hbar^2 v^2} \int_0^L \left[ \prod_{i=1}^4 dx_i \right] e^{ik_\mu(x_1 + x_2 - x_3 - x_4)}$$
$$\times \partial_{y_1,y_2}^2 \partial_{y_3,y_4}^2 \langle\langle \mathcal{G}_{he}(\boldsymbol{r}_1,\boldsymbol{r}_2) \cdot \mathcal{G}_{eh}(\boldsymbol{r}_4,\boldsymbol{r}_3) \rangle\rangle|_{y_\alpha,z_\alpha=0} \quad (21)$$

(we make explicit all of the spatial arguments in the Green's functions). On the right hand side, we replaced the average by its irreducible component; this is possible because $\langle A_h \rangle = 0$ at $k_\mu \neq 0$.

The superconductor Green's functions in Eq. (21) can be expressed in terms of the retarded Green's function $\mathcal{G}_N^R$ of the metal in the

normal state. At $E = 0$:

$$\mathcal{G}(\boldsymbol{r}_1,\boldsymbol{r}_2) = \int \frac{d\epsilon}{\pi} \frac{\epsilon \tau_z + \Delta \tau_x}{\Delta^2 + \epsilon^2} \, \text{Im} \, \mathcal{G}_N^R(\boldsymbol{r}_1,\boldsymbol{r}_2|\epsilon), \quad (22)$$

where $\tau_{x,z}$ are the Pauli matrices in the Nambu space. Substituting this relation into Eq. (21) we obtain

$$\langle |A_h|^2 \rangle = \frac{(\partial_y \Phi)^4 t^4}{\pi^2 \hbar^2 v^2} \int_0^L \left[ \prod_{i=1}^4 dx_i \right] e^{ik_\mu(x_1+x_2-x_3-x_4)}$$
$$\times \int \frac{\Delta d\epsilon}{\Delta^2 + \epsilon^2} \frac{\Delta d\epsilon'}{\Delta^2 + \epsilon'^2} \partial_{y_1,y_2}^2 \partial_{y_3,y_4}^2 \langle\langle \text{Im}\,\mathcal{G}_N^R(\boldsymbol{r}_1,\boldsymbol{r}_2|\epsilon) \cdot \text{Im}\,\mathcal{G}_N^R(\boldsymbol{r}_4,\boldsymbol{r}_3|\epsilon') \rangle\rangle|_{y_\alpha,z_\alpha=0}. \quad (23)$$

Let us focus on the averaged-over-disorder product of the Green's functions here. We can represent it as

$$\langle\langle \text{Im}\,\mathcal{G}_N^R(\boldsymbol{r}_1,\boldsymbol{r}_2|\epsilon) \cdot \text{Im}\,\mathcal{G}_N^R(\boldsymbol{r}_4,\boldsymbol{r}_3|\epsilon') \rangle\rangle$$
$$= \frac{1}{2}\text{Re}\left[ \langle\langle \mathcal{G}_N^R(\boldsymbol{r}_1,\boldsymbol{r}_2|\epsilon) \cdot \mathcal{G}_N^A(\boldsymbol{r}_4,\boldsymbol{r}_3|\epsilon') \rangle\rangle - \langle\langle \mathcal{G}_N^R(\boldsymbol{r}_1,\boldsymbol{r}_2|\epsilon) \cdot \mathcal{G}_N^R(\boldsymbol{r}_4,\boldsymbol{r}_3|\epsilon') \rangle\rangle \right], \quad (24)$$

where $\mathcal{G}_N^A$ is the advanced normal state Green's function. We will see below that the contribution of the first term to $\langle |A_h|^2 \rangle$ is determined by long diffusive electron trajectories of size $\sim \xi$. On the other hand, the contribution of the second term is determined by trajectories of length $\lesssim \lambda_F$ only ($\lambda_F$ is the Fermi wave length in the superconductor). This means that the latter contribution is small compared to the one produced by the first term in Eq. (24). In what follows we neglect the second term.

The average $\langle\langle \mathcal{G}_N^R \cdot \mathcal{G}_N^A \rangle\rangle$ can be expressed in terms of the normal-state diffuson and Cooperon[30]. Using Eq. (20) and neglecting small corrections that have a relative magnitude $\sim \lambda_F/l_{mfp} \ll 1$ (with $l_{mfp} = v_F \tau_{mfp}$ being the mean free path), we represent $\langle\langle \mathcal{G}_N^R \cdot \mathcal{G}_N^A \rangle\rangle$ as

$$\langle\langle \mathcal{G}_N^R(\boldsymbol{r}_1,\boldsymbol{r}_2|\epsilon) \cdot \mathcal{G}_N^A(\boldsymbol{r}_4,\boldsymbol{r}_3|\epsilon') \rangle\rangle$$
$$= \frac{\hbar}{2\pi \nu_M \tau_{mfp}^2} \int d^3r d^3r' \, \langle \mathcal{G}_N^R(\boldsymbol{r}_1,\boldsymbol{r}|\epsilon) \rangle \langle \mathcal{G}_N^A(\boldsymbol{r},\boldsymbol{r}_3|\epsilon') \rangle$$
$$\times \mathcal{D}_D(\boldsymbol{r},\boldsymbol{r}'|\epsilon - \epsilon') \langle \mathcal{G}_N^A(\boldsymbol{r}_4,\boldsymbol{r}'|\epsilon') \rangle \langle \mathcal{G}_N^R(\boldsymbol{r}',\boldsymbol{r}_2|\epsilon) \rangle$$
$$+ \frac{\hbar}{2\pi \nu_M \tau_{mfp}^2} \int d^3r d^3r' \, \langle \mathcal{G}_N^R(\boldsymbol{r}_1,\boldsymbol{r}|\epsilon) \rangle \langle \mathcal{G}_N^A(\boldsymbol{r},\boldsymbol{r}_4|\epsilon') \rangle$$
$$\times \mathcal{D}_C(\boldsymbol{r},\boldsymbol{r}'|\epsilon - \epsilon') \langle \mathcal{G}_N^A(\boldsymbol{r}_3,\boldsymbol{r}'|\epsilon') \rangle \langle \mathcal{G}_N^R(\boldsymbol{r}',\boldsymbol{r}_2|\epsilon) \rangle. \quad (25)$$

Here functions $\mathcal{D}_D(\boldsymbol{r},\boldsymbol{r}'|\epsilon - \epsilon')$ and $\mathcal{D}_C(\boldsymbol{r},\boldsymbol{r}'|\epsilon - \epsilon')$ are the diffuson and the Cooperon, respectively. As discussed in the beginning of the section, we focus on $B \ll H_{c2}$ and thus neglect the effect of the magnetic field penetrating the superconductor. In this case, $\mathcal{D}_D(\boldsymbol{r},\boldsymbol{r}'|\epsilon - \epsilon') = \mathcal{D}_C(\boldsymbol{r},\boldsymbol{r}'|\epsilon - \epsilon')$[30].

In the time domain, $\mathcal{D}_D(\boldsymbol{r},\boldsymbol{r}'|t)$ satisfies the diffusion equation[30],

$$(\partial_t - D\partial_{\boldsymbol{r}}^2)\mathcal{D}_D(\boldsymbol{r},\boldsymbol{r}'|t) = \delta(t)\delta(\boldsymbol{r} - \boldsymbol{r}'), \quad (26)$$

with the boundary condition corresponding to the vanishing of the probability current at the metal's surface. Here $D = v_F l_{mfp}/3$ is the diffusion constant.

At relevant energies $\epsilon - \epsilon' \sim \Delta$, the diffuson $\mathcal{D}_D(\boldsymbol{r},\boldsymbol{r}'|\epsilon - \epsilon')$ varies at a length scale of the order of $\xi$. The latter satisfies $\xi \gg l_{mfp}$ for a dirty superconductor. At the same time, the average Green's functions decay at a distance $\sim l_{mfp}$. This means that in Eq. (25) the argument $\boldsymbol{r}$ of $\mathcal{D}_D$ and $\mathcal{D}_C$ is close to $\boldsymbol{r}_1$ and the argument $\boldsymbol{r}'$ is close to $\boldsymbol{r}_2$. Consequently, we can approximate $\langle\langle \mathcal{G}_N^R \cdot \mathcal{G}_N^A \rangle\rangle$ as

$$\langle\langle \mathcal{G}_N^R(\boldsymbol{r}_1,\boldsymbol{r}_2|\epsilon) \cdot \mathcal{G}_N^A(\boldsymbol{r}_4,\boldsymbol{r}_3|\epsilon') \rangle\rangle = \frac{2\pi\nu_M}{\hbar} \mathcal{D}_D(\boldsymbol{r}_1,\boldsymbol{r}_2|\epsilon - \epsilon')$$
$$\times [V(\boldsymbol{r}_1,\boldsymbol{r}_3)V(\boldsymbol{r}_2,\boldsymbol{r}_4) + V(\boldsymbol{r}_1,\boldsymbol{r}_4)V(\boldsymbol{r}_2,\boldsymbol{r}_3)], \quad (27)$$

where we abbreviated

$$V(\mathbf{r}_1, \mathbf{r}_3) = \frac{\hbar}{2\pi\nu_M\tau_{\mathrm{mfp}}} \int d^3r \, \langle \mathcal{G}_N^R(\mathbf{r}_1, \mathbf{r}|\epsilon) \rangle \langle \mathcal{G}_N^A(\mathbf{r}, \mathbf{r}_3|\epsilon') \rangle. \tag{28}$$

Combining Eqs. (23), (24), and (27), we obtain the following expression for $\langle|A_h|^2\rangle$:

$$\begin{aligned}
\langle|A_h|^2\rangle = \frac{\nu_M(\partial_y\Phi)^4 t^4}{\pi\hbar^3 v^2} &\int_0^L \left[\prod_{i=1}^4 dx_i\right] e^{ik_\mu(x_1+x_2-x_3-x_4)} \\
&\times \int \frac{\Delta d\epsilon}{\Delta^2+\epsilon^2} \frac{\Delta d\epsilon'}{\Delta^2+\epsilon'^2} \operatorname{Re} \mathcal{D}_D(x_1, x_2|\epsilon-\epsilon') \partial^2_{y_1,y_2} \partial^2_{y_3,y_4} \\
&\times [V(\mathbf{r}_1,\mathbf{r}_3)V(\mathbf{r}_2,\mathbf{r}_4)+(3\leftrightarrow4)]|_{y_\alpha,z_\alpha=0},
\end{aligned} \tag{29}$$

where $\mathcal{D}_D(x_1, x_2|\epsilon-\epsilon') \equiv \mathcal{D}_D(\mathbf{r}_1, \mathbf{r}_2|\epsilon-\epsilon')|_{y_{1,2},z_{1,2}=0}$.

Functions $V$ in Eq. (29) stipulate $\mathbf{r}_1 \approx \mathbf{r}_3, \mathbf{r}_2 \approx \mathbf{r}_4$ in the diffuson's contribution and $\mathbf{r}_1 \approx \mathbf{r}_4, \mathbf{r}_2 \approx \mathbf{r}_3$ in the Cooperon's contribution. By making a direct calculation of the integral in Eq. (28), we find for the combination of functions $V$ in Eq. (29):

$$\begin{aligned}
&\partial^2_{y_1,y_2} \partial^2_{y_3,y_4} [V(\mathbf{r}_1,\mathbf{r}_3)V(\mathbf{r}_2,\mathbf{r}_4)+(3\leftrightarrow4)]|_{y_\alpha,z_\alpha=0} \\
&= (\pi p_F)^2 \left[\delta(x_1-x_3)\delta(x_2-x_4)+(3\leftrightarrow4)\right]/\hbar^2,
\end{aligned} \tag{30}$$

where $p_F$ is the Fermi momentum of the superconductor. The delta-functions in this expression should be interpreted as peaks of width $\sim \lambda_F$. With the help of Eq. (30), we can rewrite Eq. (29) as

$$\begin{aligned}
\langle|A_h|^2\rangle = \frac{2\pi\nu_M(\partial_y\Phi)^4 t^4 p_F^2}{\hbar^5 v^2} &\int_0^L dx_1 dx_2 \\
&\times \int \frac{\Delta d\epsilon}{\Delta^2+\epsilon^2} \frac{\Delta d\epsilon'}{\Delta^2+\epsilon'^2} \operatorname{Re} \mathcal{D}_D(x_1, x_2|\epsilon-\epsilon').
\end{aligned} \tag{31}$$

The expression for $\mathcal{D}_D(x_1, x_2|\epsilon-\epsilon')$ is sensitive to a particular geometry of the considered device. We will assume that the thickness of the superconducting film exceeds $\xi$. In this case, the film can be regarded as being three-dimensional for diffusion. We then find:

$$\mathcal{D}_D(x_1, x_2|\epsilon-\epsilon') = 2\int_0^{+\infty} \frac{dt\, e^{-i(\epsilon-\epsilon')t/\hbar}}{(4\pi Dt)^{3/2}} e^{-\frac{(x_1-x_2)^2}{4Dt}} \tag{32}$$

(the factor of 2 results from the boundary condition for Eq. (26)). Using this expression, one can easily show that

$$\int \frac{\Delta d\epsilon}{\Delta^2+\epsilon^2} \frac{\Delta d\epsilon'}{\Delta^2+\epsilon'^2} \operatorname{Re} \mathcal{D}_D(x_1, x_2|\epsilon-\epsilon') = \frac{\pi}{2D|x_1-x_2|} e^{-|x_1-x_2|/\xi}, \tag{33}$$

where $\xi = \sqrt{\hbar D/(2\Delta)}$. We will assume that the length of the proximitized segment exceeds the coherence length, $L \gg \xi$. Then, using Eq. (33) in Eq. (31) we obtain

$$\langle|A_h|^2\rangle = \frac{2\pi^2\nu_M(\partial_y\Phi)^4 t^4 p_F^2}{\hbar^5 v^2 D} \ln\frac{\xi}{l_{\mathrm{mfp}}} \cdot L. \tag{34}$$

In the latter equality, we regularized the logarithmic divergence at small distances by the mean free path $l_{\mathrm{mfp}}$, i.e., by the length scale at which the diffusive behavior ceases.

Finally, it is convenient to express the factor in front of the logarithm in Eq. (34) in terms of the normal-state conductivity of the metal $\sigma = 2e^2\nu_M D$, and of the conductance per unit length of the interface $g = 2\pi^2 G_Q t^2 (\partial_y\Phi)^2 \nu_{QH}\nu_M p_F/\hbar$. In this way we obtain Eq. (8).

## Data availability

The numerical data used to plot Fig. 1(c) are available from the corresponding author upon a reasonable request. No other data was produced.

## Code availability

The code used to produce Fig. 1(c) is available from the corresponding author upon a reasonable request. No other code was used in the study.

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

## Acknowledgements

We acknowledge very useful discussions with Ethan G. Arnault, Meng Cheng, Gleb Finkelstein, Pavel D. Kurilovich, Felix von Oppen, and Lingfei Zhao. This work was supported by NSF DMR-2002275, by Office of Naval Research (ONR) under award number N00014-22-1-2764, and by the Army Research Office (ARO) under grant number W911NF-22-1-0053 (V.D.K. and L.I.G.) and by the Yale Prize Postdoctoral Fellowship in Condensed Matter Theory (Z.M.R.).

## Author contributions

V.D.K. and L.I.G. equally contributed to the research and writing of the paper with comments, suggestions and early contributions from Z.M.R.

## Competing interests

The authors declare no competing interests.
