## [Peer Review File · Nature Communications]

Disorder-enabled Andreev reflection of a quantum Hall edgeREVIEWER COMMENTS

Reviewer #1 (Remarks to the Author):

This is a very interesting paper, which clearly deserves publication. It provides a detailed explanation of effects to be expected in a three-contact experiment where current is propagated along the edge of a quantum Hall system in contact with a grounded superconductor. The authors show how scattering due to disorder at the edge or inside the superconductor plays a crucial role in determining important statistical characteristics of measured conductance, and they are able to explain, at least qualitatively, many of the results obtained in a recent experimental paper. However, I find that the exposition is confusing in several respects, and I would ask the authors to try to clarify the presentation before the article is published. I also have some important additional questions.

The first confusing point is that the introductory paragraphs of the paper employ a semiclassical description, in which electrons in the 2DEG follow classical trajectories that undergo Andreev reflection at the boundary with the superconductor. However, the focus of the paper is on the case of a quantum Hall state at filling fraction $\nu = 2$, where electrons of both spin states are in the lowest Landau level, and a semiclassical picture is not very relevant. In fact, the calculations in the paper are fully quantum-mechanical, using the language of a chiral edge state, as is appropriate for the situation they are considering. It is not clear to me that much insight is gained by presenting a semiclassical argument, but if the authors wish to include it, they should make clear that the argument is provided as a possible motivation for their results, but that the calculations are quantum mechanical and do not depend on these arguments.

Within the semiclassical argument, it is not obvious to me why one should require that the flux through each segment of the trajectory should equal Φ_0 as stated in the caption to Fig. 1.

More generally, it seems to me that the authors' claim on page 1 that for a clean sample, electron-hole conversion is only possible for one specific value of B , is overstated. I believe that a substantial amount of conversion is possible as long as $v_F \mu$ is smaller than the pairing potential induced by the superconductor. Can the authors rule out this possibility? In the experiments of Ref. (11), where the interface is reported to have high transparency, the pairing potential should be of the order of the gap in the superconductor.

On page 2, what is the meaning of the symbol $\partial^2_{y_1 y_2}$, which first appears in Eq. (5), and then appears many times later in the article?

I do not understand the statements on page 4 that Δg and Δk_F become simple in the limit of high compressibility. Where do the associated formulas come from? Why do the authors say that high-compressibility corresponds to the limit where $\Delta \epsilon \ll \omega_c$? The center of the $\nu = 2$ Hall plateau occurs when the Fermi level is in the middle of the energy gap between the first and second Landau level. Then a high compressibility would require a high density of states in the gap, and hence a relatively dirty sample.

I found confusing the sentences following Eqs. (26) and (27), where the authors justify the result $D_D = D_C$ by saying that "a magnetic field does not penetrate a type I superconductor." In their previous discussions the authors had emphasized the importance of the superconductor being of type II, so why are they now discussing the type I case? Only much later is this answered, when the authors argue that the situation in a type II superconductor is actually not much different. It would be much better if the authors just say at the beginning that they are first ignoring phase decoherence due to the penetrating magnetic field and will later argue that the effects are small.

On page 7, line 446, the authors should say that the "thickness" of the film exceeds the penetration depth, rather than the "width".

In their discussion of the effects of the entry of vortices into the superconductor, as in Supplementary Note 3, the authors seem to assume that the vortices enter one at a time, jumping to a distance d inside the superconductor, which they estimate to be of the order of λ_A in the experiments of Ref. 11. On the other hand, one might imagine that vortex positions are correlated, and that as one vortex crosses the boundary, other vortices are moving slowly away? How would that scenario affect the authors analysis? Is there a reason for only considering one vortex at a time?

The calculations in the paper assume that there is a single chiral edge-mode for each spin, as is appropriate for $\nu = 2$, (if there is no edge reconstruction). However, the experiments of Ref. 11, which motivated the paper, include measurements of quantum Hall states up to $\nu=7$. How would the presence of multiple edge modes and the possibility of scattering between them affect the calculations in the paper? The authors should either tell us what they expect to happen when there are multiple edge modes, or they should make clear that their analysis is confined to $\nu=2$. I note that the title and abstract of the paper speak about a quantum Hall edge, without a limitation to $\nu=2$.

Reviewer #2 (Remarks to the Author):

The authors state that "Andreev reflection of an edge state is forbidden if translation invariance along the edge is preserved" and they claim that the quantization condition of the edge state at a clean NS interface cannot be satisfied except at discrete values of the magnetic field. This is mistaken. The quantization condition of figure 1 (integer number of flux quanta enclosed by the orbit) does not apply to the normal-superconductor edge, it only holds at a normal edge. There is no obstruction to Andreev reflection at a translationally invariant edge.

The authors refer to literature on Andreev edge states (Refs. 7,8), arguing that in weak magnetic fields the obstruction is ineffective, while they consider strong magnetic fields, but that is not correct. For example, PRL 98, 157003 (2007) provides a calculation of the Andreev spectrum in strong magnetic fields. The eigenstates exist at any field and evolve continuously.

This manuscript is based on a premise that is deeply flawed, I regret that I must conclude that it cannot be published.

AUTHORS' RESPONSE TO REVIEWER #1:

We are glad that the Reviewer finds our work interesting, and recommends publishing it in Nature Communications. We would also like to thank the Reviewer for the thoughtful feedback, as well as for the suggestions on how to improve the text.

Below we address all the concerns and questions raised by the Reviewer.

REVIEWER:

The first confusing point is that the introductory paragraphs of the paper employ a semiclassical description, in which electrons in the 2DEG follow classical trajectories that undergo Andreev reflection at the boundary with the superconductor. However, the focus of the paper is on the case of a quantum Hall state at filling fraction $\nu = 2$, where electrons of both spin states are in the lowest Landau level, and a semiclassical picture is not very relevant. In fact, the calculations in the paper are fully quantum-mechanical, using the language of a chiral edge state, as is appropriate for the situation they are considering. It is not clear to me that much insight is gained by presenting a semiclassical argument, but if the authors wish to include it, they should make clear that the argument is provided as a possible motivation for their results, but that the calculations are quantum mechanical and do not depend on these arguments.

REPLY:

We thank the Reviewer for bringing up this point. Indeed, the semiclassical picture – based on the notions of skipping orbits and reflection/incidence angles – does not apply to the $\nu = 2$ quantum Hall state. Therefore, we agree that over-relying on this picture may be confusing. This prompted us to deemphasize the semiclassical discussion in the introductory part.

In fact, the identified impediment to the proximitization in the translationally invariant case is quantum-mechanical in its nature. This is why it is indeed way better to discuss it in terms of the quantum Hall edge states rather than semiclassical orbits. A good segway from the semiclassical picture to a fully-quantum one is through the notion of the centers of the orbits $y_{\{c\}}$. The centers of the orbits are integrals of motion, which correspond to the edge state positions in the conditions of the quantum Hall effect. The translational invariance demands that in an Andreev reflection, the electron and hole edge state positions are symmetric with respect to the interface, $y_{\{c,n\}} + y_{\{c,m\}} = 0$. At $\nu = 2$ there is only one edge state stipulating $n = m = 1$; thus, the efficient Andreev reflection occurs if $y_{\{c,1\}} = 0$. The latter condition requires fine-tuning of, e.g., the magnetic field. We explain this in the current version of the introduction and further elaborate in the new supplement (now Supplementary Note 1).

In Supplementary Note 1, we solve the problem of transport along the proximitized, translationally invariant $\nu = 2$ edge. We demonstrate how the need for fine-tuning of the B-field comes about in the quantum-mechanical calculation. We also find the range of fields δB around the fine-tuned value B_0 in which the Andreev reflection happens with probability ~ 1 . We obtain $\delta B \sim B_0 * \Delta_{\{ind\}} / \hbar\omega_c$, where $\Delta_{\{ind\}}$ is the proximity-induced pairing potential, and ω_c is the cyclotron frequency. The ratio on the right hand side is tiny under practical conditions.

REVIEWER:

Within the semiclassical argument, it is not obvious to me why one should require that the flux through each segment of the trajectory should equal Φ_0 as stated in the caption to Fig. 1.

REPLY:

After dwelling on Reviewer #1's comment (as well as the concerns of Reviewer #2), we came to the conclusion that Fig. 1 doesn't add much to the clarity and – what's worse – causes confusion. We thus decided to take it out.

For completeness though, let us address the Reviewer's remark. In the figure, we implicitly assumed that the interface is weakly transparent, i.e., the coupling to the superconductor is weak. Under this assumption, the fluxes enclosed by electron and hole orbits are separately quantized, $\Phi_e = n\Phi_0$ and $\Phi_h = m\Phi_0$. Extending the semiclassical description to $m = n = 1$, one may see that the Andreev reflection is possible only if both orbits are semicircles with centers at the interface. This does not happen in the generic case, as we tried to illustrate with the figure.

REVIEWER:

More generally, it seems to me that the authors' claim on page 1 that for a clean sample, electron-hole conversion is only possible for one specific value of B , is overstated. I believe that a substantial amount of conversion is possible as long as $v_F \mu$ is smaller than the pairing potential induced by the superconductor. Can the authors rule out this possibility? In the experiments of Ref. (11), where the interface is reported to have high transparency, the pairing potential should be of the order of the gap in the superconductor.

REPLY:

The Reviewer is right. The Andreev reflection does indeed happen with probability ~ 1 at $|\hbar v_F \mu| < \Delta_{\text{ind}}$. The latter condition is satisfied in a narrow interval of fields around the mentioned ("fine-tuned") value B_0 , at which the edge state position $y_c = 0$. In the new supplement (Supplementary Note 1), we estimate the width of this interval as $\delta B \sim B_0 * \Delta_{\text{ind}} / \hbar \omega_c$, where ω_c is the cyclotron frequency. The ratio $\Delta_{\text{ind}} / \hbar \omega_c$ is small in practical implementations. Indeed, in fields $\sim 1\text{T}$, one can estimate $\omega_c \sim 100\text{K}$ in graphene. This exceeds by far the induced pairing potential – which is at most $\sim 10\text{K}$ in the experiment of Ref. 11.

REVIEWER:

On page 2, what is the meaning of the symbol $\partial^2_{y_1 y_2}$, which first appears in Eq. (5), and then appears many times later in the article?

REPLY:

We thank the Reviewer for pointing out that the symbol was not defined. It denotes the mixed partial derivative with respect to arguments y_1 and y_2 . In the new version, this is spelled out after Eq. (5).

The Reviewer's comment made us realize that we also didn't define the symbol ∂_y . We added the definition after Eq. (2).

REVIEWER:

I do not understand the statements on page 4 that δg and δk_μ become simple in the limit of high compressibility. Where do the associated formulas come from? Why do the authors say that high-compressibility corresponds to the limit where $\delta \epsilon \ll \omega_c$? The center of the $\nu = 2$ Hall plateau occurs when the Fermi level is in the middle of the energy gap between the first and second Landau level. Then a high compressibility would require a high density of states in the gap, and hence a relatively dirty sample.

REPLY:

Actually, in this part, we focus on relatively clean samples, in which the disorder-induced broadening of the Landau levels is small, $\delta \epsilon \ll \hbar \omega_c$. It is this condition that we use to derive δg and δk_μ . For clean samples, the chemical potential μ generically lies within one of the (broadened) Landau levels. It jumps from one broadened Landau level to another as the control parameters n and B are varied. For μ within the broadening, the compressibility is indeed high. We realized though that bringing up the compressibility is redundant. We do not mention it in the new version. In addition, we extended (what is now) Supplementary Note 3 to include the detailed derivation of δg and δk_μ .

REVIEWER:

I found confusing the sentences following Eqs. (26) and (27), where the authors justify the result $D_D = D_C$ by saying that "a magnetic field does not penetrate a type I superconductor." In their previous discussions the authors had emphasized the importance of the superconductor being of type II, so why are they now discussing the type I case? Only much later is this answered, when the authors argue that the situation in a type II superconductor is actually not much different. It would be much better if the authors just say at the beginning that they are first ignoring phase decoherence due to the penetrating magnetic field and will later argue that the effects are small.

REPLY:

We thank the Reviewer for the suggestion. In the new version, we tell from the outset that we focus on small fields, $B \ll H_{c2}$. In this case, one can neglect the effect of the field penetrating the superconductor in the derivation of $1/l_A$. We removed the mentioning of a type I superconductor, which was confusing.

REVIEWER:

On page 7, line 446, the authors should say that the "thickness" of the film exceeds the penetration depth, rather than the "width".

REPLY:

We thank the Reviewer for the suggestion, we implemented it.

REVIEWER:

In their discussion of the effects of the entry of vortices into the superconductor, as in Supplementary Note 3, the authors seem to assume that the vortices enter one at a time, jumping to a distance d inside the superconductor, which they estimate to be of the order of l_A in the experiments of Ref. 11. On the other hand, one might imagine that vortex positions are correlated, and that as one vortex crosses the boundary, other vortices are moving slowly away? How would that scenario affect the authors analysis? Is there a reason for only considering one vortex at a time?

REPLY:

Let us start by addressing the latter question. We consider the simplest case of strong pinning. Then, it is indeed reasonable to assume that vortices enter one at a time. Furthermore, vortices are pinned individually, so the entrance of a vortex does not have to displace other vortices. Due to the surface barrier for the vortex entry, its position in general is displaced from the interface by some distance, which we refer to as d . The latter is determined by combined effects of the surface barrier and pinning potential. The entered vortex abruptly changes the phase of the order parameter at the boundary, leading to the conductance jump $\delta G(d)$. In fact, such jumps were observed in the experiment of Ref. 11.

The Reviewer is right that the strong pinning is not the only possible case. In the opposite limit of weak or *collective* pinning, the vortices form a lattice weakly distorted by the pinning potential. An entering vortex would indeed displace a number of the already present ones. We conjecture that in this case a vortex entrance would still lead to a jump in the profile of the order parameter phase. However, quantitative analysis in this case requires a solution of a complex energy minimization problem for the vortex lattice, which is beyond the scope of this work.

We added a comment in the “Effect of a vortex entrance” section to make clear that we focus on the strong pinning regime.

REVIEWER:

The calculations in the paper assume that there is a single chiral edge-mode for each spin, as is appropriate for $\nu = 2$, (if there is no edge reconstruction). However, the experiments of Ref. 11, which motivated the paper, include measurements of quantum Hall states up to $\nu=7$. How would the presence of multiple edge modes and the possibility of scattering between them affect the calculations in the paper? The authors should either tell us what they expect to happen when there are multiple edge modes, or they should make clear that their analysis is confined to $\nu=2$. I note that the title and abstract of the paper speak about a quantum Hall edge, without a limitation to $\nu=2$.

REPLY:

We thank the Reviewer for the question. We think that the generalization of our basic findings to $\nu > 2$ is straightforward. We expect that the conductance G of a long proximitized edge remains a random quantity with $\langle G \rangle = 0$ and a symmetric with respect to $G = 0$ distribution

function. The conductance variance can be estimated with the help of the random matrix theory (see, e.g., Phys. Rev. B 83, 085413); we find $\langle\langle G^2 \rangle\rangle \sim G_Q^2$ regardless of ν . The independence of $\langle\langle G^2 \rangle\rangle$ of ν is in the spirit of the universal conductance fluctuations. We leave the generalization of the more sophisticated predictions (such as those related to the correlation function of G) to a future work.

To address the Reviewer's concern, we added a paragraph to the Discussion section, in which we comment on the case of $\nu > 2$.

AUTHORS' RESPONSE TO REVIEWER #2:

We appreciate the feedback of the Reviewer. The Reviewer's comments made us realize that our presentation was confusing in several respects. We believe that the confusing presentation led the Reviewer to an **incorrect** conclusion that our theory is flawed. We hope that our replies, together with the significant modifications of the text, resolve the misunderstanding.

REVIEWER:

The authors state that "Andreev reflection of an edge state is forbidden if translation invariance along the edge is preserved" and they claim that the quantization condition of the edge state at a clean NS interface cannot be satisfied except at discrete values of the magnetic field. This is mistaken. The quantization condition of figure 1 (integer number of flux quanta enclosed by the orbit) does not apply to the normal-superconductor edge, it only holds at a normal edge.

REPLY:

After thinking carefully about the Reviewer's comment, we have to admit that Fig. 1 was extremely confusing (although not wrong). It heavily relied on an assumption – which we made only implicitly – that the interface is weakly-transparent. (For a weakly-transparent interface, the electron and hole orbits are indeed quantized separately via $\Phi = n\Phi_0$). We did not clarify this in the text or the figure caption. This led to the Reviewer's misunderstanding and, eventually, contributed to their conclusion that our theory is flawed. We will explain below that in fact there is no flaw in our theory. To resolve the confusion, we removed Fig. 1 and added a supplement (Supplementary Note 1) to the new version.

REVIEWER:

There is no obstruction to Andreev reflection at a translationally invariant edge.

REPLY:

We agree with the Reviewer: formally, there is no obstruction to Andreev reflection at a translationally invariant quantum Hall edge. Some amount of particle-hole conversion remains there at any magnetic field B . However, we claim that the Andreev reflection occurs with probability ~ 1 only in a **narrow** interval of B around a certain field value B_0 .

The width of the interval can be estimated as $\delta B \sim B_0 * \Delta_{\text{ind}} / (\hbar\omega_c) \ll B_0$ in the realistic regime of $\Delta_{\text{ind}} \ll \hbar\omega_c$, where ω_c is the cyclotron frequency (this relation holds even if the induced pairing Δ_{ind} reaches the value of gap in the superconductor). Away from this narrow interval – i.e., **for the majority of the field values** – the probability of the Andreev reflection is small ($\ll 1$). This is what we meant by "**generically** the Andreev reflection of an edge state is forbidden if translational invariance along the edge is preserved." In the new version, we replaced the word "forbidden" by "suppressed".

To make the origin of suppression clear, we added a new supplement (now Supplementary Note 1). In this supplement, we solve a fully quantum-mechanical problem of transport along the

proximitized, translationally invariant edge. We demonstrate that the probability of the Andreev reflection is indeed generically small, except for a narrow interval δB . We also derive the above mentioned estimate and provide the characteristic value for δB .

REVIEWER:

The authors refer to literature on Andreev edge states (Refs. 7,8), arguing that in weak magnetic fields the obstruction is ineffective, while they consider strong magnetic fields, but that is not correct. For example, PRL 98, 157003 (2007) provides a calculation of the Andreev spectrum in strong magnetic fields. The eigenstates exist at any field and evolve continuously.

REPLY:

We agree with the Reviewer: the eigenstates do indeed exist at any field B and evolve continuously with it. However, it is the **particle-hole content of the eigenstates** that determines the probability of the Andreev reflection. At a certain ("fine-tuned") field value B_0 particle and hole components have the same weight. Then, and only then, the Andreev reflection happens with a probability ~ 1 . The probability drops off fast, once the field is outside a narrow vicinity of B_0 .

In more detail, at each energy value, the BdG equations have two solutions. If the field is in the vicinity of B_0 , then the particle and hole amplitudes are of the same order in each of the two eigenstates. Outside that vicinity, one of the eigenstates is predominantly particle-like, and another one is hole-like. Therefore, the probability of the Andreev reflection becomes small once B leaves the vicinity (of width δB) of B_0 . The estimate for the width is $\delta B \sim B_0 * \Delta_{\text{ind}} / \hbar\omega_c$. For the practical conditions $\Delta_{\text{ind}} \ll \hbar\omega_c$. This leads to a stringent condition on the field required for the efficient Andreev reflection, $\delta B/B_0 \ll 1$. (We note in passing that PRL 98, 157003 (2007) focuses on the opposite limit of $\Delta_{\text{ind}} \gg \hbar\omega_c$, as illustrated in its Fig. 2.)

REVIEWER:

This manuscript is based on a premise that is deeply flawed, I regret that I must conclude that it cannot be published.

REPLY:

We respectfully disagree with the Reviewer that the premise of our manuscript is flawed. We believe that the judgment of the Reviewer is a result of a confusion. We hope that we managed to resolve it by our reply. We also significantly modified the manuscript in line with the reply.

SUMMARY OF CHANGES:

1. We changed the word “forbidden” to “suppressed” in the abstract.
2. We modified the introductory part, which explains that disorder is needed for an efficient Andreev reflection. Previously, the explanation relied on quasiclassical analysis. In the current version, we present a fully quantum-mechanical argument.
3. We added a new supplementary note (now Supplementary Note 1), in which we present a comprehensive solution to the problem of transport along the proximitized, translationally invariant quantum Hall edge.
4. We removed Fig. 1 which caused confusion to both Reviewers.
5. We defined the symbol ∂_y after Eq. (2).
6. We defined the symbol $\partial^2_{y1, y2}$ after Eq. (5).
7. We modified (what is now) Supplementary Note 3. Now it includes the derivation of the expressions for $\delta g/g$ and δk_μ presented in the in-line equations of the main text. The supplementary note includes a new figure (Fig. 1).
8. We found a typo in the expression for $\delta g/g$ and fixed it. The correct expression reads $\delta g/g = \delta B/B$, instead of $2\delta B/B$.
9. We added a comment to section “Effect of a vortex entrance”, in which we explain the presented treatment is valid under the assumption of strong pinning.
10. We added a paragraph to the Discussion section, in which we consider the case of $\nu > 2$.
11. We modified the beginning of the Methods section following the suggestion of the Reviewer #1. We removed the mentioning of a type I superconductor, which was confusing. In the new version, we tell from the outset that our approach applies to a type II superconductor in small fields, $B \ll H_{c2}$.

REVIEWER COMMENTS

Reviewer #1 (Remarks to the Author):

The authors have adequately responded to the criticisms and suggestions in my earlier report, and I am pleased to recommend publication. I have only two minor suggestions.

The authors should point out that equation S1 neglects the effects of Zeeman coupling, which would lead to a difference in the Fermi wave vectors of the two spin species. This neglect is presumably justified to the extent that the Zeeman splitting is small compared to the induced superconducting gap.

On line 346 of the main text, I believe that the first formula gives the value of $(\Delta \theta)^2$, rather than $\Delta \theta$. This will not affect the estimate of T_{\sim} .

Reviewer #2 (Remarks to the Author):

I regret that I still find the analysis deeply confusing. The authors state as their central premise that "generically Andreev reflection of an edge state is suppressed if translation invariance along the edge is preserved." They state that Andreev reflection needs a "fine tuning" of the magnetic field to be "on resonance".

These words, and the qualitative description in the intro, paint a picture that is at odds with the formulas. Eq. S2 in the supplemental shows that Andreev reflection decays quadratically as $(\lambda/\xi)^2$, with $\lambda=1/k$ the Fermi wave length and $\xi=\hbar v/\Delta$ the superconducting coherence length. For a "resonance" I would have expected some commensurability relation, this decay looks more like a wavelength mismatch or impedance mismatch. For such a smooth dependence on the parameters words like "fine tuning" and "suppression" seem an exaggeration.

I also take issue with the statement that this "suppression" of Andreev reflection is "generic". It only holds for a weak tunnel coupling with the superconductor, when this coupling can be treated perturbatively. The experiments which the authors wish to explain claim to have a high transparency. Why is the theory still applicable?

AUTHORS' RESPONSE TO REVIEWER #1:

We thank the Reviewer for their recommendation to publish our work. We also appreciate their two suggestions, which are both on point.

REVIEWER:

The authors should point out that equation S1 neglects the effects of Zeeman coupling, which would lead to a difference in the Fermi wave vectors of the two spin species. This neglect is presumably justified to the extent that the Zeeman splitting is small compared to the induced superconducting gap.

REPLY:

We thank the Reviewer for bringing this point up. Indeed, we neglect the Zeeman splitting in Eq. (S1). We clarify this after the equation in the new version of the text.

We note that the Zeeman effect would not qualitatively change the conclusions of Supplementary Note 1. One can straightforwardly show that Eq. (S2) remains valid even in the presence of the Zeeman splitting; the only modification is that k_{μ} gets replaced by $(k_{\mu\uparrow} + k_{\mu\downarrow}) / 2$. Accordingly, an appreciable particle-hole conversion happens only at a single value of the field at which $k_{\mu\uparrow} = -k_{\mu\downarrow}$. Away from this field value, the Andreev reflection is suppressed.

REVIEWER:

On line 346 of the main text, I believe that the first formula gives the value of $(\delta \theta)^2$, rather than $\delta \theta$. This will not affect the estimate of T_{\sim} .

REPLY:

We thank the Reviewer for finding the typo, we fixed it in the new version.

AUTHORS' RESPONSE TO REVIEWER #2:

We thank the Reviewer for their feedback.

REVIEWER:

I regret that I still find the analysis deeply confusing. The authors state as their central premise that "generically Andreev reflection of an edge state is suppressed if translation invariance along the edge is preserved." They state that Andreev reflection needs a "fine tuning" of the magnetic field to be "on resonance".

These words, and the qualitative description in the intro, paint a picture that is at odds with the formulas. Eq. S2 in the supplemental shows that Andreev reflection decays quadratically as $(\lambda/\xi)^2$, with $\lambda=1/k$ the Fermi wave length and $\xi=\hbar v/\Delta$ the superconducting coherence length. For a "resonance" I would have expected some commensurability relation, this decay looks more like a wavelength mismatch or impedance mismatch. For such a smooth dependence on the parameters words like "fine tuning" and "suppression" seem an exaggeration.

REPLY:

1. We would prefer to keep the notions of "fine tuning" and "suppression" in the paper. There is no ambiguity as to what these terms mean in the main text. Furthermore, the terms are quantified in the Supplementary Information.

Specifically, in Supplementary Note 1 we show that the probability of the particle-hole conversion is **minuscule** unless the magnetic field B is tuned to a **narrow** vicinity δB of field B_0 . Field B_0 corresponds to the edge state position $y_c = 0$ with respect to the interface; B_0 is of the order of the field required for reaching the $\nu = 2$ quantum Hall state. The presented analytical result – together with the numerical estimates for the parameters of the experiment [Nature Physics 16, 862 (2020)] – yields $\delta B/B_0 \sim 0.1$, clearly justifying the use of the terms "fine tuning" and "suppression".

2. We used the word "resonant" once, in the Supplementary Note 1. Our motivation came from the similarity of the coefficient of the sine in Eq. (S2) to the Breit-Wigner formula. With that said, we have to admit that the use of this term may be confusing (no standing waves can form at a chiral edge, so the notion of resonant transmission does not apply). Therefore, we agree with the Reviewer and remove the reference to the resonance from the text.

REVIEWER:

I also take issue with the statement that this "suppression" of Andreev reflection is "generic". It only holds for a weak tunnel coupling with the superconductor, when this coupling can be treated perturbatively. The experiments which the authors wish to explain claim to have a high transparency. Why is the theory still applicable?

REPLY:

We firmly believe that our theory is directly applicable to the experiment [Nature Physics 16, 862 (2020)].

1. In the SI, we remove the constraint on the strength of tunnel coupling and solve the particle-hole conversion problem in the absence of disorder. Specifically, we consider the edge state propagation along the boundary that includes a finite proximitized segment of length L ; the induced superconducting pairing potential Δ_{ind} of arbitrary strength turns on and off sharply at the ends of the segment. The found conversion probability is

$$p \sim \frac{(\Delta_{\text{ind}}/\hbar v k_{\mu})^2}{1 + (\Delta_{\text{ind}}/\hbar v k_{\mu})^2}, \quad (1)$$

cf. the second term in Eq. (S2).

The induced pairing potential depends on the transmission coefficient, but can not exceed the magnitude of the parent gap Δ even at high transparency of the graphene-superconductor interface. We will optimistically assume that the proximitization in [Nature Physics 16, 862 (2020)] is ideal. In this case, $\Delta_{\text{ind}} \sim \Delta$, where $\Delta \sim 10\text{K}$ is the gap in the MoRe superconductor.

The edge state velocity v depends on a number of material parameters. Its numerical evaluation performed specifically for the conditions of experiment was included in [Nature Physics 16, 862 (2020)]. The Authors of that publication conclude that their “simulation suggests that the CAES velocity is probably lower but comparable to v_F ” (here v_F is the Fermi velocity in graphene, and CAES stands for chiral Andreev edge states). We use this result of a published work to replace v with v_F in Eq. (1). Then, for the generic position of the chemical potential, the product $\hbar v_F k_{\mu}$ is of the order of the gap $\hbar \omega_c$ between the Landau levels, $\hbar v_F k_{\mu} \sim \hbar \omega_c$. In graphene, $\hbar \omega_c \sim 100\text{K}$ in fields of the order of a few Tesla. Combining the estimates, we find

$$p \sim (\Delta / \hbar \omega_c)^2 \sim 0.01 \ll 1.$$

The low value of p verifies that in the absence of disorder the particle-hole conversion remains suppressed, even in the limit of high transparency of the interface. (We added the above estimate to Supplementary Note 1.)

2. The derived above estimate of p is the most generous one. The value of p is highly sensitive to whether the pairing potential Δ_{ind} turns on and off abruptly or gradually at the ends of the proximitized region. In the latter case, the propagation of the edge states is adiabatic. According to the general adiabatic principle, the eigenstates preserve their nature upon a slow variation of the parameters, i.e., the incoming particle (hole) exits the proximitized region as a particle (hole), see arXiv:2210.08048 for a specific example. Therefore, the conversion probability $p \rightarrow 0$ **regardless** of the interface transparency (including the high interface transparency limit brought up by the Reviewer).

The particle-hole conversion occurs due to the violation of adiabaticity by the imperfections. In the conditions of the experiment [Nature Physics 16, 862 (2020)], the mentioned violation likely originates from the disorder in the superconductor (to reach a high H_{c2} , a “dirty” superconductor was used). Our theory provides concrete predictions for the statistical properties

of the particle-hole conversion in precisely this, practical case. Therefore, we are of the firm opinion that our work is directly applicable to the experiment [Nature Physics 16, 862 (2020)].

We believe that points 1 and 2 fully resolve the Reviewer's concern regarding the applicability of our theory to the experiment.

SUMMARY OF CHANGES:

1. We fixed a typo in section “Conductance fluctuations at finite temperature” by changing $\delta\theta$ to $(\delta\theta)^2$.
2. We defined the velocity v after Eq. (S1) in Supplementary Note 1.
3. We added a comment saying that we disregard the Zeeman effect in Eq. (S1).
4. We removed the word “resonant” from Supplementary Note 1.
5. We added an estimate for the particle-hole conversion probability in the absence of disorder in Supplementary Note 1.
6. We added a footnote commenting on the edge state velocity at the NS interface in Supplementary Note 1.

REVIEWERS' COMMENTS

Reviewer #2 (Remarks to the Author):

I do not object to publication. It's a fine piece of theoretical work, applicable to a recent experiment.

AUTHORS' RESPONSE TO THE SECOND REVIEWER

REVIEWER:

I do not object to publication. It's a fine piece of theoretical work, applicable to a recent experiment.

REPLY:

We thank the reviewer for their assessment.

SUMMARY OF CHANGES:

- As a final change, we added a parenthetical remark after Eq. (10) which we forgot to include in the previous version ["(we also promoted $\alpha(x)$ and $\theta(x)$ from the variables...)"]. This change is optional, but it should benefit the clarity of the text.